# Structural basis for DNA 5′-end resection by RecJ

**Kaiying Cheng, Hong Xu, Xuanyi Chen, Liangyan Wang, Bing Tian, Ye Zhao\*, Yuejin Hua\***

Key Laboratory of Chinese Ministry of Agriculture for Nuclear-Agricultural Sciences, Institute of Nuclear-Agricultural Sciences, Zhejiang University, Hangzhou, China

**Abstract** The resection of DNA strand with a 5′ end at double-strand breaks is an essential step in recombinational DNA repair. RecJ, a member of DHH family proteins, is the only 5′ nuclease involved in the RecF recombination pathway. Here, we report the crystal structures of *Deinococcus radiodurans* RecJ in complex with deoxythymidine monophosphate (dTMP), ssDNA, the C-terminal region of single-stranded DNA-binding protein (SSB-Ct) and a mechanistic insight into the RecF pathway. A terminal 5′-phosphate-binding pocket above the active site determines the 5′-3′ polarity of the deoxy-exonuclease of RecJ; a helical gateway at the entrance to the active site admits ssDNA only; and the continuous stacking interactions between protein and nine nucleotides ensure the processive end resection. The active site of RecJ in the N-terminal domain contains two divalent cations that coordinate the nucleophilic water. The ssDNA makes a 180° turn at the scissile phosphate. The C-terminal domain of RecJ binds the SSB-Ct, which explains how RecJ and SSB work together to efficiently process broken DNA ends for homologous recombination.

## Introduction

DNA double-strand breaks (DSBs) are the most lethal form of DNA damage due to the free DNA ends. DSBs can be repaired by homologous recombination (HR), which requires a pre-aligned homologous sequence prior to ligation. DNA end resection degrades the 5′ strand of a DSB end and is one of the earliest and most important processes in HR repair (*Symington and Gautier, 2011*). In bacteria DSBs are predominantly repaired by either the RecBCD or the RecF pathways. Compared with the RecBCD pathway, the RecF pathway is highly conserved across eubacteria and shows functional homology in the eukaryotic HR process. For example, RecA resembles yeast Rad51 protein with regard to the catalysis of DNA strand invasion and change reactions (*Cox and Lehman, 1982*). RecO is capable of annealing single-stranded DNA-binding (SSB) protein-coated ssDNA and facilitates RecA loading (*Kantake et al., 2002*), a property that is shared with the yeast Rad52 protein. The RecBC complex contains both helicase and nuclease activities. In contrast, the RecF pathway requires RecJ, a 5′-3′ exonuclease, together with RecQ helicase and SSB protein to initiate DSB end resection (*Courcelle and Hanawalt, 1999*; *Han et al., 2006*; *Handa et al., 2009*; *Morimatsu and Kowalczykowski, 2014*).

*Deinococcus radiodurans* is extremely resistant to DNA-damaging agents, such as ionizing radiation, ultraviolet radiation and mitomycin C (MMC). This robustness correlates with its extraordinary DNA repair capabilities, especially DSB repair, which can rebuild a shattered genome within several hours (*Krisko and Radman, 2013*; *Slade et al., 2009*). In contrast to *Escherichia coli*, RecBC is naturally absent from *D. radiodurans*, whereas the key components of the RecF pathway are present. Disruption of *recF, recO,* or *recR* significantly sensitized the cells to radiation similar as *recA* mutant (*Bentchikou et al., 2010*; *Ithurbide et al., 2015*; *Satoh et al., 2012*; *Xu et al., 2008*), which suggests that the RecF pathway is the dominant HR pathway in *D. radiodurans*. An extended synthesis-

\*For correspondence: yezhao@
zju.edu.cn (YZ); yjhua@zju.edu.cn
(YH)

**Competing interests:** The
authors declare that no
competing interests exist.

**Reviewing editor:** Stephen C
Kowalczykowski, University of
California, Davis, United States

**eLife digest** DNA encodes information that cells need to create the molecules and proteins that are essential for life. It is therefore vital that damaged DNA is repaired rapidly and accurately. Some DNA-damaging agents, such as gamma radiation, break both strands of the DNA double helix, which can be fatal to cells if not repaired quickly and accurately.

One important pathway in charge of repairing such double-strand breaks is called the homologous recombination repair pathway. The first stage of this repair involves cutting away part of one of the DNA strands at the break. This exposes a single-stranded stretch of the partner strand, which can be used for the repair.

One organism that is highly resistant to having its DNA damaged by radiation is the bacterium *Deinococcus radiodurans*. In this bacterium, an enzyme called RecJ performs part of the first step in the repair of DNA double-strand breaks by progressively shortening one end of a DNA strand. Cheng et al. have now used crystallography to look at the structure that RecJ forms when it binds to DNA. This, together with the results from biochemical experiments, revealed how RecJ recognizes where to bind on a broken DNA strand and how it moves along the broken strand along with cutting that strand.

Further investigations revealed that two other proteins enhance the ability of RecJ to process the ends of broken DNA strands. Cheng et al. also examined the structure that RecJ forms with one of these additional proteins, called SSB. A future goal is to determine how all three proteins co-ordinate with each other to efficiently and accurately repair double stranded breaks in the *D. radiodurans* bacteria.

dependent strand annealing process prior to DNA recombination contributes substantially to the rapid restoration of an intact genome (*Bentchikou et al., 2010*; *Slade et al., 2009*). It was proposed that RecJ nuclease, which is associated with helicase, rapidly digests the DSB end in a 5′-3′ polarity after DNA damage. The resultant 3′-ssDNA overhang subsequently anneals to the complementary DNA strand via RecA- and RadA-mediated strand invasion, followed by extensive DNA synthesis.

RecJ orthologs have been found widespread in most eubacteria and archaea (*Aravind and Koonin, 1998*; *Sanchez-Pulido and Ponting, 2011*). RecJ was firstly identified in *E. coli* by its effect on recombination that cells lacking both RecBCD and RecJ showed extreme recombination deficiency (*Lovett and Clark, 1984*). In addition to HR, RecJ is involved in ssDNA gap repair, base excision repair and methyl-directed mismatch repair (*Burdett et al., 2001*; *Dianov et al., 1994*; *Lovett and Kolodner, 1989*). As a processive nuclease, RecJ only degrades ssDNA in a 5′-3′ direction but not capable of DNA end digestion with a blunt end or 3′-ssDNA overhang (*Cheng et al., 2015b*; *Han et al., 2006*; *Jiao et al., 2012*). Notably, RecJ also showed 'dRPase' activity to incise abasic sites (*Dianov et al., 1994*). SSB can interact with RecJ and stimulate its DNA binding and nuclease activities (*Han et al., 2006*; *Morimatsu and Kowalczykowski, 2014*; *Sharma and Rao, 2009*). Recently, it has been shown that the initiation of DSB end resection by coordinated activities of RecJ and RecQ depends on the nature of the DNA ends (*Morimatsu and Kowalczykowski, 2014*). In *E. coli*, RecJ nuclease alone is capable of digesting DNA with 5′-ssDNA overhang, whereas RecQ, a 3′-5′ helicase, is required to initiate DNA end resection with a blunt end or 3′-ssDNA overhangs (*Morimatsu and Kowalczykowski, 2014*).

The crystal structures of RecJ orthologue from *Thermus thermophilus* (ttRecJ) have been solved in the absence of substrate DNA (*Wakamatsu et al., 2010*; *Yamagata et al., 2002*). The structure of ttRecJ reveals an O-shaped molecule, which comprises a novel oligonucleotide/oligosaccharide-binding (OB) fold domain. The hole close to the active site is too narrow to accommodate double-strand DNA (dsDNA), suggesting the ssDNA preference of ttRecJ. The nuclease core of ttRecJ consists of a central parallel β-sheet encircled by α-helices. Two catalytic metal ions coordinated by conserved histidine and aspartate residues were observed in the active site. However, in the absence of DNA substrate, the conserved loop region of the nuclease core was disordered.

Here, we report three crystal structures of *D. radiodurans* RecJ (drRecJ) in complex with deoxythymidine monophosphate (dTMP), ssDNA, and the C-terminal region of drSSB protein (SSB-Ct).

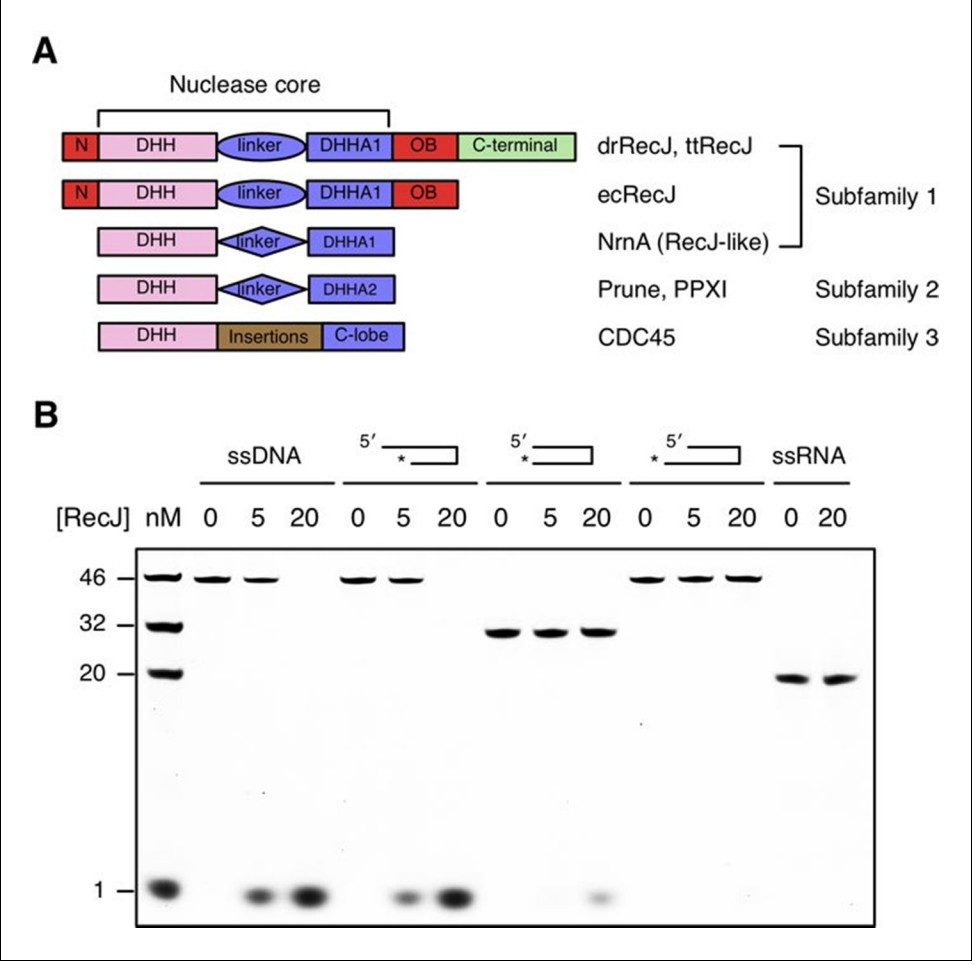

**Figure 1.** Domain arrangement and substrate specificity of drRecJ. (**A**) Schematic of the domain arrangements of three DHH subfamilies. (**B**) Denaturing PAGE gel showing that drRecJ degrades different substrates, as shown at the top of the panel. 3'-Fluorescent labeled DNA or RNA (100 nM) were incubated with drRecJ (0, 5 and 20 nM) in the presence of 100 nM $Mn^{2+}$ (see methods).

The following figure supplements are available for figure 1:

**Figure supplement 1.** Sequence alignments, secondary structure, and functional residues of RecJ and CDC45.

**Figure supplement 2.** Metal preference of drRecJ.

These structures, together with mutagenesis and biochemical studies, provide mechanistic insights into DNA resection by drRecJ.

## Results

### Protein characterization and crystallization

DHH family proteins can be divided into three subfamilies based on their sequence similarity and domain arrangement (*Figure 1A*). These subfamilies share a conserved N-terminal DHH domain that consists of consecutive DHH residues, which give rise to the name this family of proteins. The subfamily 1 group includes RecJ and nanoRNase (RecJ-like protein), a nuclease that degrades short RNA, and has a distinct DHHA1 domain following the DHH domain (*Srivastav et al., 2014*; *Wakamatsu et al., 2010*). In contrast, the DHHA2 domain is present in the subfamily 2 group (e.g., exopolyphosphatase PPX1 and *Drosophila* Prune protein) (*Ugochukwu et al., 2007*). The subfamily

3 group is defined by the RecJ eukaryotic orthologue CDC45, which has large insertions between the DHH domain and the C-lobe (*Krastanova et al., 2012*). Notably, bacterial RecJ nucleases have an additional OB fold domain next to the nuclease core (*Figure 1A*).

drRecJ is conserved within bacteria, which shares 42% and 32% amino acid identity with ttRecJ and ecRecJ, respectively (*Figure 1—figure supplement 1*). The sequence alignment revealed that drRecJ contains all the seven signature motifs of the RecJ family nucleases, which belongs to the DHH (motifs I-V) domain and the DHHA1 domain (motifs VI and VII) (*Figure 1—figure supplement 1*). Compared with ecRecJ, drRecJ has an additional C-terminal domain that is conserved in the *Deinococcus-Thermus* phylum (*Figure 1A* and *Figure 1—figure supplement 1*). To determine the metal preference, various concentrations of $Mn^{2+}$ and $Mg^{2+}$ were applied to reactions (*Figure 1—figure supplement 2*). In contrast to the degradation of DNA by ecRecJ in a reaction that requires $Mg^{2+}$, both $Mn^{2+}$ and $Mg^{2+}$ can activate the nuclease activity of drRecJ. However, the drRecJ activity in a reaction buffer containing 10 µM $Mn^{2+}$ is substantially higher than that contains 10 mM $Mg^{2+}$ (*Figure 1—figure supplement 2*). Overwhelming $Mg^{2+}$ (10 mM) does not affect the nuclease activity when drRecJ is pre-incubated with $Mn^{2+}$ (0.1 mM), which suggests that $Mn^{2+}$ ions are employed for drRecJ catalysis. To test the substrate specificity, drRecJ was incubated with different types of synthetic DNA fluorescence-labeled at the 3′ end in the presence of $Mn^{2+}$ (*Figure 1B*). drRecJ is able to processively digest ssDNA and DNA with a 5′-ssDNA overhang (14 nt overhang) to a single nucleotide (*Figure 1B*). In contrast, drRecJ cannot resect ssRNA or DNA with a blunt end or 3′-ssDNA overhang (6 nt overhang), which indicates that a free 5′-ssDNA is essential for drRecJ nuclease activity.

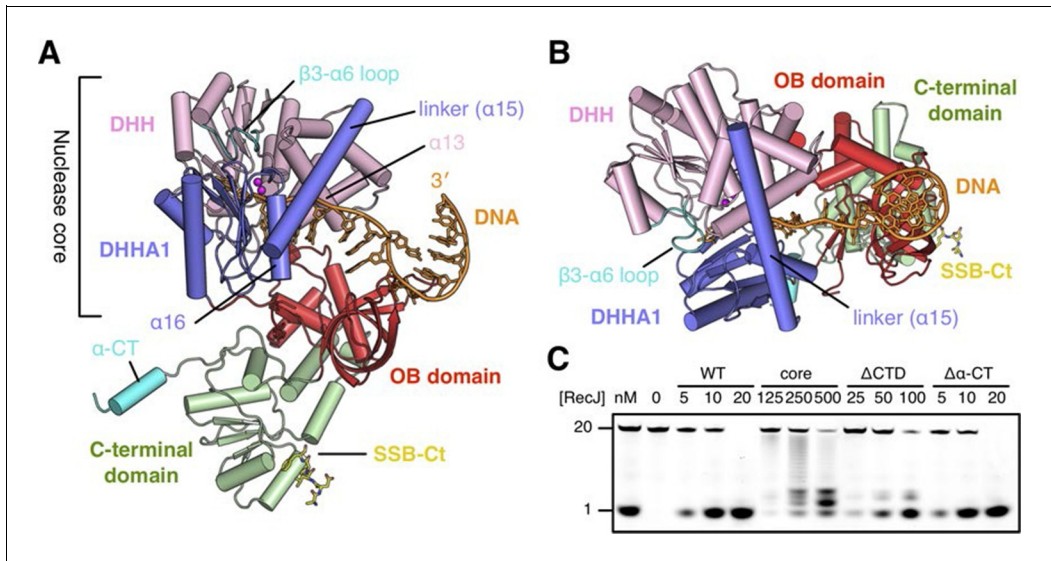

**Figure 2.** Structure of drRecJ complex. (**A**) Overall structure of drRecJ complex viewed from the side. Protein domains of drRecJ are labeled and shown in distinct colors. The DNA and SSB-Ct are colored orange and yellow, respectively. Two $Mn^{2+}$ in the active site are shown as magenta spheres. Two regions that are disordered in the ttRecJ structures (PDB code: 2ZXP) are highlighted in cyan. Three helices that form a helical gateway are also labeled. (**B**) Overall structure of the drRecJ complex viewed from the top of the DNA. The downstream nucleotides stack well to mimic the double-stranded DNA. (**C**) Denaturing PAGE gel showing the nuclease activities of different truncations of drRecJ. 3′-Fluorescence-labeled 20 nt ssDNA (100 nM) was incubated with various concentrations of different truncations of drRecJ proteins (see methods).

The following figure supplements are available for figure 2:

**Figure supplement 1.** Structure of RecJ:DNA complex.

**Figure supplement 2.** Comparison of the nuclease fold and DHHA1/DHHA2 domain.

**Figure supplement 3.** Relative position of DHH (pink) and DHHA1/DHHA2 (blue) domain.

**Table 1.** Statistics from crystallographic analysis.

| | RecJ-dTMP | RecJ$_d$-DNA | RecJ$_d$-DNA-SSBct |
|---|---|---|---|
| | Complex I | Complex II | Complex III |
| **Data collection** | | | |
| Space group | $P\,3_2 21$ | $P\,3_2 21$ | $P\,3_2 21$ |
| Cell dimensions a, b, c (Å) | 106.53 | 105.83 | 102.22 |
| | 106.53 | 105.82 | 102.22 |
| | 161.90 | 165.40 | 166.12 |
| Wavelength (Å) | 0.9792 | 0.9792 | 0.9792 |
| Resolution (Å) | 30–2.7 (2.77–2.70) | 30–2.6 (2.66–2.60) | 30–2.3 (2.35–2.30) |
| R-meas | 5.5 (64.1) | 6.5 (79.1) | 7.4 (63.7) |
| I/σI | 27.0 (4.0) | 22.6 (3.0) | 17.0 (2.9) |
| Completeness (%) | 98.9 (99.5) | 99.6 (99.4) | 99.4 (94.1) |
| Redundancy | 8.9 | 8.3 | 7.2 |
| | | | |
| **Refinement** | | | |
| Resolution (Å) | 30–2.7 | 30–2.6 | 30–2.3 |
| No. reflections | 29747 | 33594 | 45296 |
| $R_{work}$/$R_{free}$ | 20.61/25.20 | 18.47/22.84 | 22.56/23.89 |
| No. atoms | | | |
| Protein/DNA | 5262/- | 5373/286 | 5342/182 |
| Ligand/Ion | 21/2 | 35/2 | 55/2 |
| Waters | 12 | - | 142 |
| B factors | | | |
| Protein/DNA | 79.8/- | 70.4/103.9 | 58.2/93.7 |
| Ligand/Ion | 66.7/76.6 | 99.8/68.2 | 74.5/61.9 |
| Water | 62.7 | - | 48.9 |
| Rmsd | | | |
| Bond length (Å) | 0.012 | 0.005 | 0.012 |
| Bond Angle (°) | 1.085 | 0.849 | 1.464 |
| Ramachandran statitics | | | |
| Favored (%) | 98.00 | 99.1 | 98.9 |
| Allowed (%) | 2.0 | 0.9 | 1.1 |
| Outliers (%) | 0 | 0 | 0 |

Values in parentheses refer to the highest resolution shell.

R factor = $\Sigma||F(obs)- F(calc)||/\Sigma|F(obs)|$.

Rfree = R factor calculated using 5.0% of the reflection data randomly chosen and omitted from the start of refinement.

RecJ$_d$ denotes catalytic inactive drRecJ (H160A)

To characterize the RecJ recognition and incision of the DNA substrate, we crystallized three types of drRecJ complexes: wild-type drRecJ complexed with dTMP (complex I), a catalytic inactive mutant (H160A) RecJ complexed with DNA bearing a 5′-ssDNA overhang (complex II), and the ternary complex of RecJ-ssDNA and the SSB-Ct (complex III). The crystals were grown in the presence of $Mn^{2+}$ ions and diffracted X-rays to 2.3–2.7 Å resolution. The structures are validated by the appearance of a well-formed active site with two catalytic metal ions and DNA or dTMP at the active

site (*Figure 2A,B*). The crystal data, together with the data collection and refinement statistics, are summarized in *Table 1*.

## Architecture of the RecJ nuclease

drRecJ contains four domains: an N-terminal DHH domain (residues 49–295), a DHHA1 domain (residues 328–424), an OB fold domain (residues 1–48 and residues 425–532), and an extended C-terminal domain (residues 533–705; *Figure 2A–B* and *Figure 1—figure supplement 1*). The DHH domain and DHHA1 domain are interconnected by a long linker α-helix (α15; residues 296–327) to form the nuclease core, as previously observed for the structure of ttRecJ (*Yamagata et al., 2002*). The overall structure and conformation of the drRecJ complexes can be virtually superimposed on the ttRecJ (PDB code: 2ZXP) with the rmsd value of 1.937–2.307 Å over 476 Cα atoms (*Figure 2—figure supplement 1A*). Although the DHHA1 domain and OB fold domain can be adequately superimposed, the DHH domain and linker α-helix further shift towards the DHHA1 domain, resulting in a much narrower cleft that accommodates the substrate DNA (*Figure 2—figure supplement 1A*). The C-terminal domain shows large movement relative to the nuclease core, which is most likely due to the crystal lattice contacts. Three loops, which are located at the OB fold domain, also show noticeable deviations due to interactions with the substrate DNA binding (*Figure 2—figure supplement 1A*). Above the active site, the β3-α6 loop, which is disordered in the ttRecJ structures, caps the DNA (*Figure 2A*). The C-terminal-most α-helix (α-CT) also becomes ordered in all the drRecJ structures (*Figure 2A*).

Recognition of the ssDNA (complex II and III) by the drRecJ is mediated by nuclease core and OB fold domain (*Figure 2A–B* and *Figure 2—figure supplement 1B*). The DHHA1 domain and the DHH domain bind the 5′-upstream region of the ssDNA. The two-metal ion active site in the DHH domain stabilizes the scissile phosphate. Three helices α13, α15 and α16 form a helical gateway, which only permits ssDNA passage. The OB fold domain, located directly at the side of the DNA entrance, interacts with the downstream region of the ssDNA. The C-terminal domain, in contrast, is involved in SSB-Ct binding. These DNA binding properties were confirmed by mutagenesis coupled with nuclease assays (*Figure 2C*). Compared with wild-type protein, the nuclease core of drRecJ exhibits a much decreased nuclease activity and processivity on ssDNA, which indicates that the OB fold domain is critical for substrate DNA binding. However, drRecJ lacking the C-terminal domain (ΔCTD) also exhibits reduced nuclease activity, which suggests that the C-terminal domain is possibly involved in OB fold domain orientation.

As the representative member of the DHH family proteins, the DHH domain of drRecJ consists of α/β repeats, in which five central parallel β-strands (β2 to β6 in drRecJ) are surrounded by α-helices (*Figure 2—figure supplement 2A*). The order of the parallel β-strands is 21345, which is shared by all the DHH family proteins (e.g., NrnA in *Figure 2—figure supplement 2A*) (*Uemura et al., 2013*; *Ugochukwu et al., 2007*; *Yamagata et al., 2002*). The signature DHH residues (motif III) are located at the end of the fourth strand. The residues at the end of the first and third β-strands and the DHH motif coordinate two catalytic metal ions to form the active site (*Figure 2—figure supplement 2A*). The arrangement of the catalytic residues appears to be employed by many other nucleases (*Figure 2—figure supplement 2A*). For example, despite the different topology of the β-strands (32145), the catalytic residues of human flap endonuclease 1 (hFen1) are also located between the end of three central parallel β-strands (*Tsutakawa et al., 2011*). As noted above, subfamily 1 and subfamily 2 DHH family proteins have distinct domains next to the DHH domain, which are denoted DHHA1 and DHHA2, respectively (*Figure 1A*). Both the DHHA1 domain and the DHHA2 domain are structurally similar with a mixed five-stranded central β-sheet surrounded by α-helices (*Figure 2—figure supplement 2B*). The topologies of the β-strands, however, are different from DHHA1 (12354, ↑↑↓↑↓) and DHHA2 (12345, ↑↓↓↑↓). In addition, the β1-β2 and β3-β4 strands in the DHHA1 domain have two inserted α-helices (A and B in RecJ and NrnA, respectively, in *Figure 2—figure supplement 2B*); in the DHHA2 domain, the α-helices are located between the β2-β3 and β4-β5 strands (bottom two panels in *Figure 2—figure supplement 2B*). In drRecJ, the first (A) α-helix (α16) of DHHA1 is critical to the composition of the helical gateway (*Figure 2A*), which may explain the varying substrate specificity between these two subfamilies.

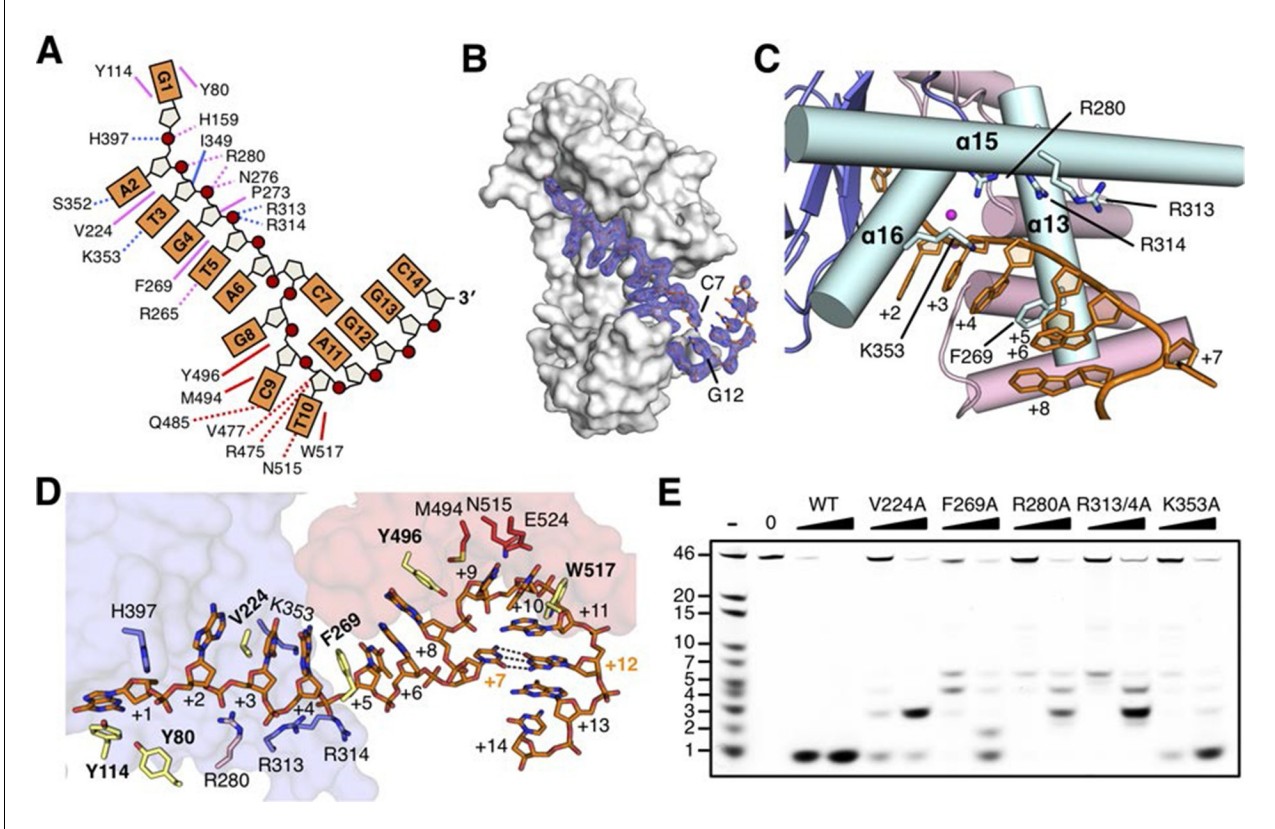

**Figure 3.** The DNA binding in the RecJ-DNA complex. (**A**) Schematic of the numbered DNA substrate used for complex II crystallization. Interactions between nucleotides and DHH domain, DHHA1 domain and OB fold domain are colored pink, blue and red, respectively. Hydrogen bonds are defined as within 3.2 Å and van der Waals contacts within 4.2 Å (dashed lines). Solid lines indicate residues that stack with DNA bases. (**B**) drRecJ surface and $2F_{o-}F_c$ electron density of DNA contoured at 1σ. The C7-G12 base pair is labeled. (**C**) The helical gateway is labeled and shown in cyan. Key residues interacting with DNA are labeled and shown as sticks. Nucleotides are labeled as in (**A**). (**D**) Interactions between drRecJ and DNA in complex II structure. Protein side chains involved in the protein-DNA interactions are shown as sticks, and the key residues that form stacking interactions are highlighted in yellow. Nucleotides are labeled as in (**A**). C7 and G12 (orange) form Watson-Crick base pair, as indicated by the dark dashed line. (**E**) Denaturing PAGE gel showing the reduced nuclease activity and processivity of mutant drRecJ proteins (alanine substitutions of key residues involved in DNA binding). 3'-Fluorescence-labeled 46 nt ssDNA (100 nM) was incubated with drRecJ proteins (0, 5 and 20 nM) in the presence of 100 nM $Mn^{2+}$ (see methods).

The following figure supplements are available for figure 3:

**Figure supplement 1.** The DNA binding of drRecJ.

**Figure supplement 2.** DNA binding activity of drRecJ mutant proteins as in the *Figure 3E*.

## Helical gateway and stacking interactions between DNA and RecJ nuclease

drRecJ primarily contacts DNA with helix and loop elements (*Figure 3* and *Figure 3—figure supplement 1A*). In both complex II and complex III structures, a positively charged groove between the DHH domain and the DHHA1 domain was observed to bind 5 nt 5′-upstream ssDNA in the same manner (*Figure 3A* and *Figure 3—figure supplement 1B*). Multiple loops located at the interface between the DHH domain and the DHHA1 domain bind the first two nucleotides (*Figure 3—figure supplement 1A*). In contrast, the helical gateway, which is composed of three α-helices, primarily binds the +3 to +5 nucleotides (*Figure 3C,D*). Ala substitutions of key residues in the helical gateway (Arg280 in α13; Arg313/Arg314 in α15, and Lys353 in α16) reduce the DNA-binding and nuclease activity with multi-stops at 5–7 nt (*Figure 3E* and *Figure 3—figure supplement 2*), which suggests that the helical gateway is essential for DNA binding and translocation. The interactions between

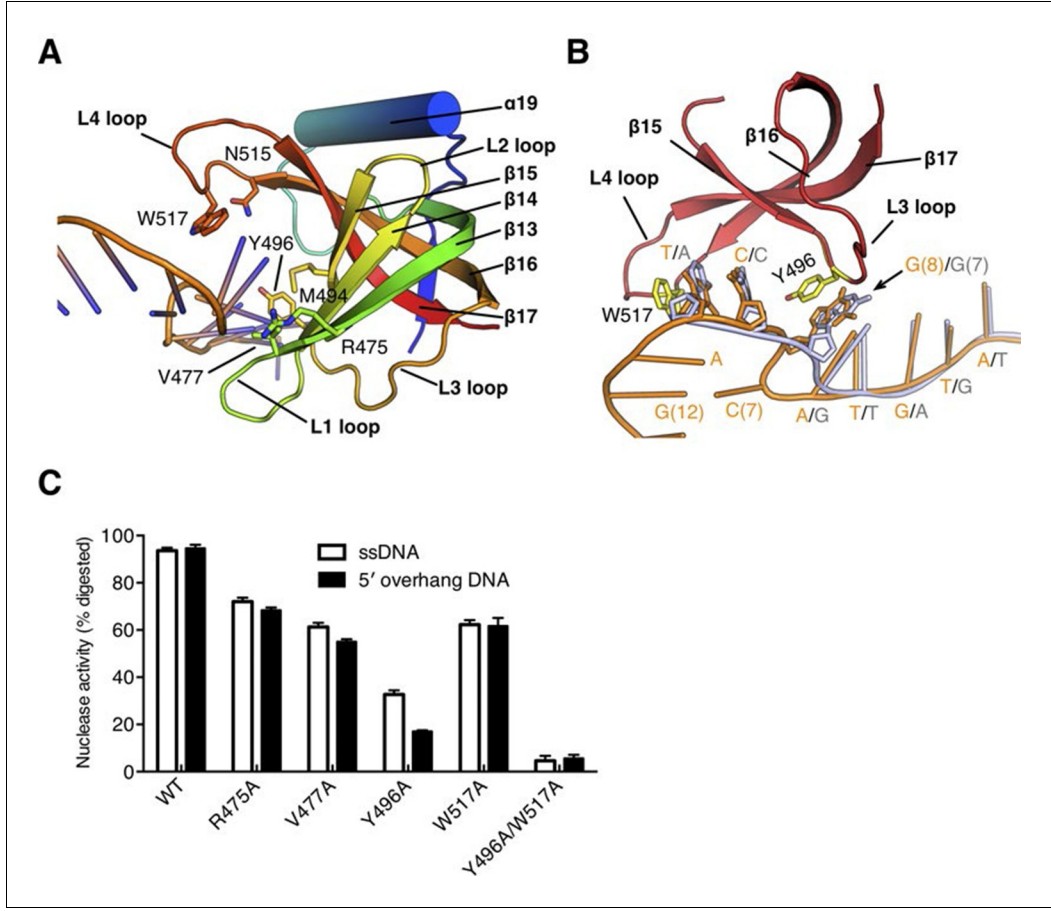

**Figure 4.** The OB fold domain is critical for drRecJ resection. (**A**) The OB fold domain is shown in rainbow-colored diagrams. Residues involved in DNA binding are labeled and shown as sticks. (**B**) A comparison of DNA in the complex II (DNA with 5´-ssDNA overhang; orange) with complex III (ssDNA; white). Two conserved aromatic residues Tyr496 and Trp517 are shown as sticks (yellow). The black arrowhead indicates the position of stacking interaction between Tyr496 and the guanine base. (**C**) Quantification and plot of ssDNA and DNA with 5´-ssDNA overhang, which are processed by wild-type and mutant drRecJ proteins (alanine substitutions of key residues in the OB fold domain), using the same DNA substrate and reaction conditions as in *Figure 1B*. Data are represented as mean ± SEM.

The following figure supplement is available for figure 4:

**Figure supplement 1.** Structural comparison of the OB fold from *E. coli* SSB (ecSSB), *D. radiodurans* SSB subunits (drSSB_NTD and drSSB_CTD), *D. radiodurans* RecO (drRecO) and human replication protein A subunits (RPA14, RPA32 and RPA70).

the nuclease core and DNA phosphate groups are primarily formed by a number of Arg, His, and Asn residues (*Figure 3A,D*). Notably, four side chain residues form stacking interactions to the 5´-upstream DNA bases (*Figure 3D*). Tyr114 and Tyr80 (motif I) interact with the +1 base at the active site. The substitution of deoxyribose sugar with ribose sugar clashes with the Tyr114 residue, which is consistent with the notion that drRecJ nuclease only acts on DNA (*Figure 1B*). Val224 (motif IV) and Phe269 (motif V) insert between the +2/+3 base and +4/+5 base. Ala substitutions of these two residues also impair the DNA-binding, nuclease activity and processivity of drRecJ (*Figure 3E* and *Figure 3—figure supplement 2*).

The OB fold domain of drRecJ at the side of the DNA entrance interacts with three downstream nucleotides (+8 to +10; *Figure 3A,D*). This domain comprises an N-terminal region and a typical OB fold located next to the DHHA1 domain (*Figure 2A*). Five β-strands (β13 to β17) in the OB fold are orthogonally located to form a mixed β-barrel (*Figure 4A*), which has a Greek topology (12354) that

is identical to the topology of other proteins (*Figure 4—figure supplement 1*) (*Bernstein et al., 2004*; *Bochkareva et al., 2002*; *Leiros et al., 2005*; *Raghunathan et al., 2000*). Compared with the canonical OB fold involved in ssDNA binding, the OB fold of drRecJ exhibits certain discrepancies. The L2 loop (β14-β15 loop) is short and the top of the β-barrel is capped by an additional α helix (*Figure 4A* and *Figure 4—figure supplement 1*). Conversely, the bottom α helix that connects the L3 loop (β15-β16 loop) is absent in drRecJ (*Figure 4A* and *Figure 4—figure supplement 1*). Interestingly, a similar structural features were observed in the previously solved drRecO structure (*Leiros et al., 2005*) (*Figure 4—figure supplement 1E*), suggesting the possible coevolution of the RecF pathway. Surprisingly, no direct interaction between the side chain residues and the DNA phosphate group was observed (*Figure 3A,D*). Two conserved aromatic residues, Tyr496 and Trp517 from the L3 and the L4 loop (β16-β17 loop), with residues (Met494, Arg475 and Val477) between the end of the first three β-strands form the downstream DNA-binding surface, which interacts with the nucleotide bases (*Figure 4A,B*). Ala substitutions of these residues reduce the nuclease activity of drRecJ on both ssDNA and DNA with 5′-ssDNA overhangs (*Figure 4C*). Notably, Y496A mutant protein exhibits more severely impaired nuclease activity on DNA bearing 5′-ssDNA overhang compared with ssDNA (*Figure 4C*), suggesting that the OB fold is critical for the drRecJ resection of DNA with 5′-ssDNA overhang.

Despite the disorder of the last four nucleotides, the G12, G13 and C14 in complex II structure stack well, which mimic the double stranded DNA (*Figure 2B* and *Figure 3D*). Unexpectedly, while drRecJ continuously interacts with the ssDNA (complex III structure, *Figure 4B*), the C7 base in the 5′-ssDNA overhang of complex II is flipped out into the solvent (*Figure 3D* and *Figure 4B*). The flipped out cytosine forms the Watson-Crick base pair with G12, which causes severe bending at approximately 100° at the ss-dsDNA junction (*Figure 2A*, *Figure 3D* and *Figure 4B*). This base flipping is most likely attributed to the stacking interaction between Tyr496 and the purine base (G8 in complex II and G7 in complex III; *Figure 4B*). Poly(dA) and poly(dT) oligomers were synthesized to confirm the interaction between Tyr496 and purine base. Indeed, wild-type drRecJ exhibits approximate 2-fold increase in the catalytic efficiency ($k_{cat}/K_m$) for the poly(dT) substrate compared to the poly(dA) substrate (*Table 2*). When Tyr496 is replaced by Ala, the mutant drRecJ has a reduced $K_m$ and $k_{cat}$ but is no longer sensitive to the substrate sequence context, confirming the preferred stacking interaction between Tyr496 and purine base.

## Nucleotide binding site and the catalytic mechanism

In complex I (drRecJ-dTMP) structure, the dTMP is located at the interface between the DHH domain and the DHHA1 domain (*Figure 5A*). The thymine base and the deoxyribose are coordinated by Tyr80, Tyr114 and two β-strands (β10 and β11) from the DHHA1 domain (*Figure 5A,B*). The phosphate group of the dTMP is held in place by Arg109, Arg280 (motif V), Ser371 and Arg373

**Table 2.** Kinetic parameters of wild-type and mutant drRecJ proteins.

| Protein-substrate | $K_m$ (nM) | $k_{cat}$ (min$^{-1}$) | $k_{cat}/K_m$ (µM$^{-1}$ min$^{-1}$) |
| --- | --- | --- | --- |
| WT-KY09 (poly(dT)) | 74.9 ± 7.6 | 1.61 ± 0.03 | 21.5 |
| WT-KY08 (poly(dA)) | 100.3 ± 5.5 | 1.15 ± 0.02 | 11.5 |
| Y496A -KY09 (poly(dT)) | 102.2 ± 10.9 | 0.32 ± 0.01 | 3.1 |
| Y496A -KY08 (poly(dA)) | 109.1 ± 8.4 | 0.27 ± 0.01 | 2.5 |
| WT-KY03 | 90.3 ± 4.5 | 1.55 ± 0.01 | 17.1 |
| Y80A-KY03 | 223.1 ± 12.2 | 0.25 ± 0.02 | 1.1 |
| Y114A-KY03 | 349.4 ± 23.8 | 0.77 ± 0.05 | 2.2 |
| R109A-KY03 | 158.3 ± 19.9 | 0.25 ± 0.02 | 1.6 |
| S371A-KY03 | 104.4 ± 8.8 | 0.78 ± 0.03 | 7.5 |
| R373A-KY03 | 349.6 ± 26.2 | 0.61 ± 0.03 | 1.7 |
| R393A-KY03 | 105.8 ± 6.9 | 1.51 ± 0.02 | 14.2 |
| H397A-KY03 | 321.3 ± 22.0 | 0.77 ± 0.03 | 2.4 |

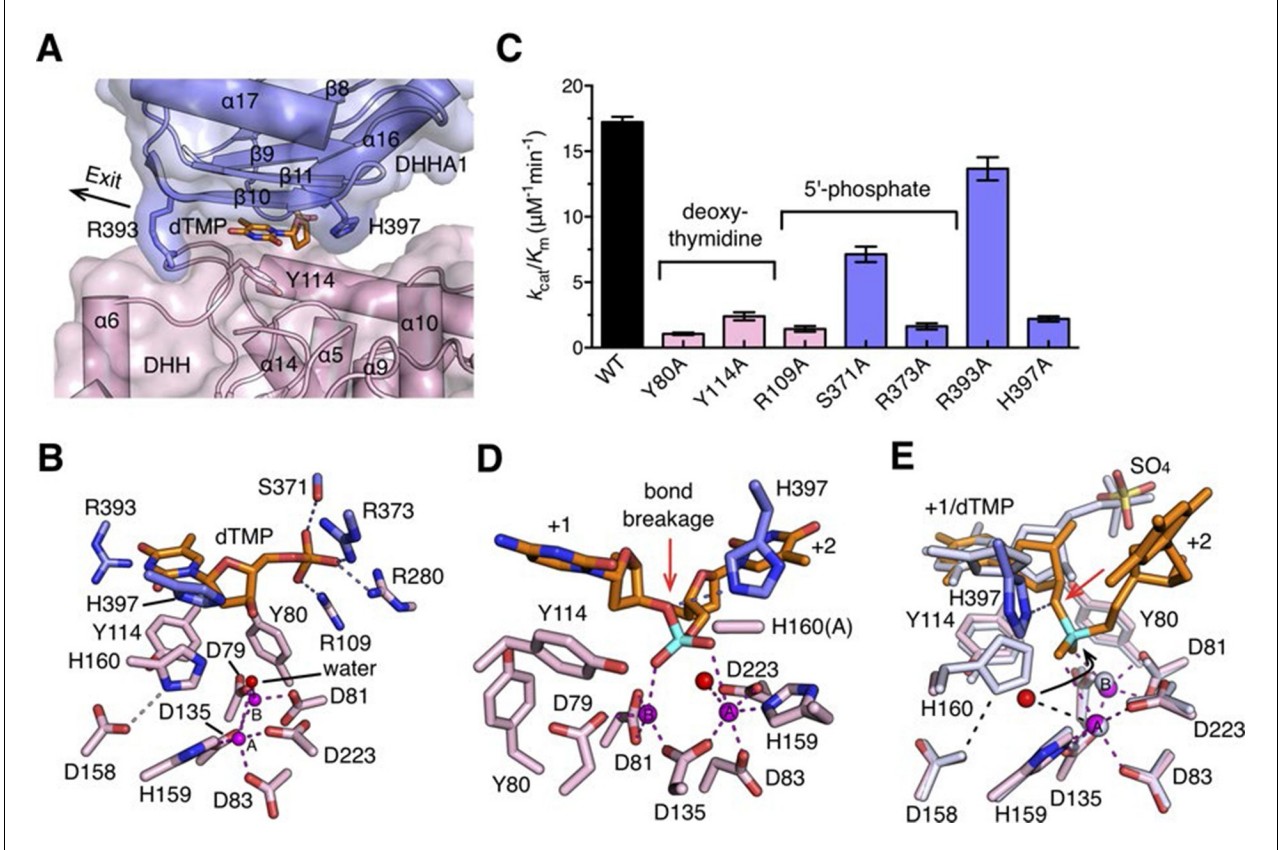

**Figure 5.** Nucleotide binding site and the catalysis. (A) dTMP binding pocket between the DHH domain and DHHA1 domain. dTMP is shown as stick (orange). Two conserved residues His397 and Arg393 at the entrance and exit to the active site are labeled and shown as sticks. (B) Close-up of the active site of drRecJ-dTMP (complex I). Two $Mn^{2+}$ ions (A and B; magenta) are coordinated by a water molecule (red sphere) and conserved Asp and His residues (magenta dashed lines). Asp158 forms a hydrogen bond with His160, as indicated by the pink dashed line. The phosphate of the dTMP is held by Arg109, Arg280, Ser371 and Arg373 (blue dashed lines). (C) Catalytic efficiency of wild-type and mutant drRecJ as a bar graph showing the relative severity of the mutations. The Michaelis-Menten kinetics data are from *Table 2*. Data are represented as mean ± SEM. (D) Close-up of the active of site of complex III. The scissile phosphate centered on two catalytic $Mn^{2+}$ is highlighted in cyan. The red arrowhead indicates the position of P-O bond breakage. Interaction between His397 and the oxygen atom of the scissile phosphate group is indicated by blue dashed line. (E) A comparison of the active sites in the complex I (white) with complex III (same color as in panel D). The nucleophilic water molecule observed in complex III occupies the position close to the His160 in drRecJ-dTMP structure (complex I). The red arrowhead indicates the position of P-O bond breakage and the black arrowhead indicates the direction of nucleophilic attack.

The following figure supplement is available for figure 5:

**Figure supplement 1.** Denaturing PAGE gel showing the inactivation of mutant drRecJ proteins (alanine substitutions of residues involved in metal-ion-chelation).

(motif VI; *Figure 5B*). This mononucleotide-binding pocket is situated immediately above the active site, which is consistent with the translocation of the DNA by one nucleotide required for the exonu-clease activity to proceed. Mutations of the binding pocket dramatically impair the enzymatic activity (*Figure 5C* and *Table 2*). The catalytic center of the complex I consists of two metal ions and two bridging ligands, a water molecule and the side chain of Asp135 (motif II; *Figure 5B*). These two metal ions are bound by five aspartate residues and one histidine: one metal ion (A) is coordinated by Asp83 (motif I), Asp135, Asp223 (motif IV) and His159 (motif III), whereas Asp79, Asp81 (motif I) and Asp135 are the ligands to the other metal ion (B; *Figure 5C*). Alanine substitutions of these metal-binding residues caused an almost complete inactivation of drRecJ (*Figure 5—figure supple-ment 1*), which is consistent with the results of equivalent mutations in ecRecJ (*Sutera et al., 1999*). In addition, two conserved positively charged residues (His397 and Arg393) from motif VII are

situated at the entrance and exit to the active site (*Figure 5A,B*). Alanine substitution of Arg393 shows a modest effect on drRecJ catalytic efficiency.

The active site of the drRecJ-DNA can be virtually superimposed well with that of the drRecJ-dTMP (*Figure 5D,E*). The DNA scissile phosphate in drRecJ-DNA is centered on two $Mn^{2+}$ ions and the A site metal ion seems to be in a position on the nucleophile side (*Figure 5D*). The deoxyribonucleioside in complex I occupies almost the same place as the +1 nucleotide observed in drRecJ-DNA (*Figure 5E*). Despite of the slightly movements, the coordination of the two catalytic metal ions is almost identical, except for an additional coordination between the A site metal ion and a water molecule (*Figure 5D,E*). This water molecule is located directly below the scissile phosphate at a distance of 3.3 Å and is a probable nucleophile candidate for the nucleophilic attack. However, the angle between the water, scissile phosphate and 3′-O leaving group is ~115 degree (*Figure 5E*), which explains the inactivity of the drRecJ-DNA complex. This DNA conformation is possibly attributed to the Ala substitution of His160, which most likely serves as the general base essential for catalysis (*Figure 5E*). Alanine substitutions of the conserved His160 and Asp158, which is hydrogen bonded to His160 (*Figure 5B,E*), inactivate drRecJ nuclease activity (*Figure 5—figure supplement 1*), which suggests that the DHH motif is critical for the catalysis. In addition, His397 from β10 undergoes a substantial rotamer change during DNA binding (*Figure 5E*). This residue, which is located opposite to the direction of the nucleophilic attack in drRecJ-DNA complex, is hydrogen bonded to the oxygen atom of the scissile phosphate group (*Figure 5D,E*). Alanine substitution of His397 drastically reduced the rate of the drRecJ-catalyzed reaction (*Figure 5C*), suggesting that this residue may serve as a general acid to protonate the 3′-O leaving group.

## The C-terminal domain of RecJ is critical for protein-protein interaction

Inspection of the complex III electron density map reveals an additional electron density corresponding to the C-termini of the SSB-Ct peptide associated with the drRecJ C-terminal domain (*Figure 6A*). Clear electron density was observed as the last four residues of SSB-Ct peptide (residues 298–301, ED<u>DLPF</u>) used for co-crystallization. In contrast, the first two residues (Glu296 and Asp297) of the SSB-Ct appear to be disordered. Three helices α20, α21 and α23 form the SSB-Ct peptide-binding site, which anchors the last SSB-Ct residue (Phe301) in a deep hydrophobic pocket (*Figure 6A*). The aromatic Phe side chain is packed against Pro553, Pro628 and the side chains of Met557, Val572, Tyr575 and Tyr600 from drRecJ (*Figure 6B*). Moreover, Asp298 of SSB-Ct forms an apparent ionic bond with the Lys629 side chain, which composes an electropositive patch near the N-terminus of SSB-Ct (*Figure 6B* and *Figure 3—figure supplement 1B*). These SSB-Ct binding site features are similar to those found in the ribonuclease HI SSB-Ct binding site and appears to be shared by other SSB partner proteins (*Figure 6—figure supplement 2*) (*Marceau et al., 2011*; *Petzold et al., 2015*; *Ryzhikov et al., 2011*). Y575A mutant drRecJ (RecJ$_{575}$), wild-type drSSB (SSB$_{WT}$) and drSSB, which lacks eight C-terminal residues (SSB$_{ΔC}$), were purified to confirm the interactions between the SSB-Ct domain and the C-terminal domain (*Figure 6C* and *Figure 6—figure supplement 1*). drRecJ or RecJ$_{575}$ alone can not process DNA with a 3′-ssDNA overhang (*Figure 6C*, lanes 2–3 and 8–9), which is consistent with previous biochemical observations of ecRecJ (*Morimatsu and Kowalczykowski, 2014*). For wild-type drRecJ, this nuclease activity is enhanced by SSB$_{WT}$ (lanes 4–5) but not by SSB$_{ΔC}$ (lanes 6–7). In contrast, neither SSB$_{WT}$ (lanes 10–11) nor SSB$_{ΔC}$ (lanes 12–13) stimulate the RecJ$_{575}$ resection of DNA with a 3′-ssDNA overhang substantially, indicating that the interactions between drRecJ and drSSB are required for the stimulation of drRecJ resection of DNA with a 3′-ssDNA overhang. For ssDNA, the stimulation is similar to that of the resection of the DNA with a 3′-ssDNA overhang, suggesting that the SSB-RecJ interaction is required for RecJ to resect SSB-covered ssDNA (*Figure 6—figure supplement 1*).

The α-CT is located outside the C-terminal domain and exposed to solvent, which has no effect on drRecJ nuclease activity (*Figure 2C*). To test whether the α-CT is required for drRecJ function in vivo, we performed the phenotypic assays (*Figure 7A*). Cells that lack *recJ* (Δ*recJ*) are thermosensitive and sensitive to MMC treatment (*Figure 7A*). Interestingly, while complementation of the entire RecJ in Δ*recJ* strain (Δ*recJ/pk-recJ*) completely recovered the WT phenotype, overexpression of RecJ that lacks α-CT (Δ*recJ/pk-recJΔCα*), only partially recovered the phenotype, suggesting the essential role of the α-CT for drRecJ function in vivo (*Figure 7A*). It has been shown that RecQ and RecJ biochemically cooperate to process DNA with 3′-ssDNA overhang in *E. coli* (*Morimatsu and Kowalczykowski, 2014*). However, drRecQ is able to activate both wild-type drRecJ and drRecJ

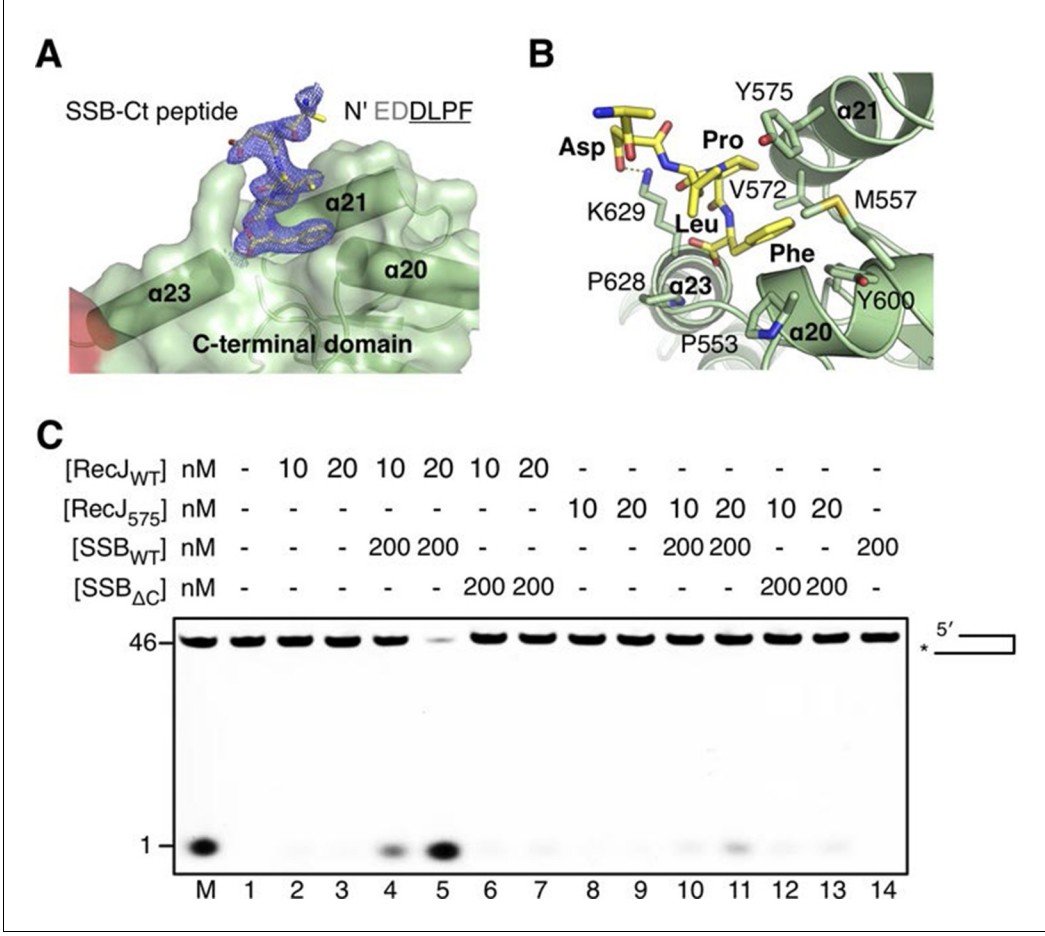

**Figure 6.** The C-terminal domain interacts with the SSB-Ct. (**A**) SSB-Ct binding pocket. SSB-Ct is shown as stick. The electron density of SSB-Ct is shown in blue with the refined 2Fo-Fc map contoured at 1σ. (**B**) Interactions of the SSB-Ct and C-terminal domain. Both SSB-Ct (yellow) and residues involved in SSB-Ct interactions (green) are shown as sticks. The ionic bond between the Asp298 of SSB-Ct and Lys629 of drRecJ is indicated by a yellow dashed line. (**C**) SSB enhances wild-type drRecJ degradation. Y575A mutant drRecJ (RecJ575), wild-type drSSB (SSB_WT) and drSSB lacking eight C-terminal residues (SSB_ΔC) were purified to perform the nuclease assays. For the reaction, DNA with a 3´-ssDNA overhang was pre-incubated with 200 nM SSB_WT or SSB_ΔC and treated with drRecJ or RecJ575 (5 and 20 nM) (see Materials and methods).

The following figure supplements are available for figure 6:

**Figure supplement 1.** SSB enhances wild-type drRecJ degradation on ssDNA.

**Figure supplement 2.** Structural comparison of the SSB-Ct binding pockets of RecJ, Rnase HI (4Z0U), RecO (3Q8D) and Pol III (3SXU).

lacking α-CT resection of DNA with 3´-ssDNA overhang (unpublished data) similarly, suggesting that the α-CT is possibly involved in the interaction between drRecJ and other helicases in vivo. Moreover, although both drSSB and drRecQ can enhance drRecJ nuclease activity on DNA with 3´-ssDNA overhang, the resection is substantially stimulated when both drRecQ and drSSB were present (*Figure 7B*), which suggests that the DNA with 3´-ssDNA overhang is most likely processed by the combined activities of RecJ, RecQ and SSB in vivo (*Figure 7C*).

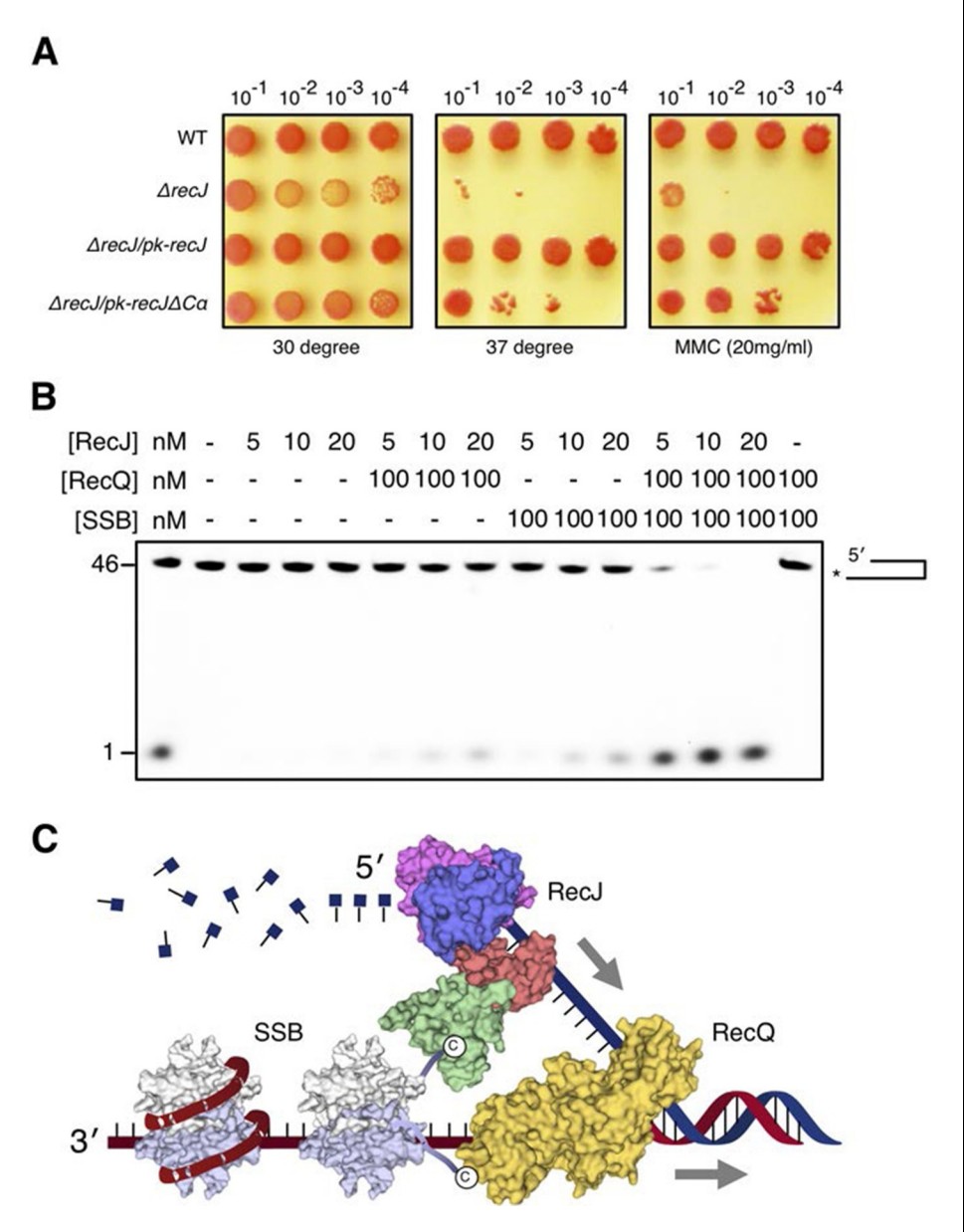

**Figure 7.** DSB end resection requires the coordinate activities of RecJ, RecQ and SSB proteins. (**A**) Functional analysis of the drRecJ α-CT in vivo. Wild-type (WT), *recJ* mutant (Δ*recJ*) and *recJ* complementary (Δ*recJ/pk-recJ* for the entire RecJ and Δ*recJ/pk-recJΔCα* for RecJ lacking α-CT) strains were spotted on TGY medium following high-temperature (37 degrees) and MMC treatments. (**B**) drRecJ processes DNA bearing 3´-ssDNA overhang together with drRecQ and drSSB. The reaction contained drRecJ (5, 10 and 20 nM), drRecQ (100 nM) and drSSB (100 nM). (**C**) A model for DSB end resection by RecJ, RecQ and SSB proteins in *D. radiodurans*. RecQ (yellow) is bound to the ss-dsDNA junction, which unwinds them to generate 5´-tailed ssDNA. Following RecJ digestion, the SSB (homodimer, white and grey) is recruited to the resultant 3´-ssDNA overhang, which facilitates further strand exchange reaction.

## Discussion

### Unified understanding of DHH family members

DHH family proteins have been found widespread in bacteria, archaea and eukaryotes, which contain five conserved motifs. Analyses of the conserved DHH domain and the nonconserved DHHA1

domain in drRecJ structures provide insight into the conserved catalysis mechanism but different substrate specificity of these proteins. As a family of proteins, they share the conserved topology of β-strands in the DHH domain, the signature DHH motif at the end of the fourth strand and the catalytic residues (*Figure 2—figure supplement 2A*). The core of the DHH domain, which consists of a central five parallel β-strands sandwiched by surrounding α helices, appears to be shared by many other nucleases. The similar Rossmann-like arrangement of the active site suggests that these nucleases may employ similar two-metal-ion catalysis (*Figure 2—figure supplement 2A*).

The DHHA1 domain in the subfamily 1 group, however, shows distinct features from the DHHA2 domain in the subfamily 2 group. In addition to the different topology of the β-strands, the subfamily 2 group lacks the helical gateway element (α16 inserted between the first two β-strands in DHHA1) as observed in drRecJ. More importantly, the DHH domain of the subfamily 1 group faces the C-terminal three β-strands of DHHA1 to form the DNA binding site (*Figure 2—figure supplement 3A*), whereas the active site of the subfamily 2 group is formed between the DHH domain and the first β-strand of DHHA2 (*Figure 2—figure supplement 3B*). In addition, the C-terminal α-helix of DHHA2 caps the active site, resulting in a much narrower cleft that only accommodates the small-size pyrophosphate or polyphosphate. Thus, the relative positions of the DHH and the DHHA1/DHHA2 domain may determine the substrate specificity between these two subfamily groups.

CDC45, which is in the subfamily 3 group, is a eukaryotic orthologue of RecJ with no nucleolytic cleavage activity but retaining ssDNA binding capability (*Krastanova et al., 2012*; *Petojevic et al., 2015*; *Szambowska et al., 2014*). In eukaryotes, CDC45, the Mcm2-7 helicase and GINS proteins form the CMG complex, which is involved in DNA replication (*Miller et al., 2014*; *Pacek et al., 2006*). Recent studies revealed that CDC45 may guard the gate of Mcm2/5 helicase and capture the errant leading strand (*Petojevic et al., 2015*). Based on sequence alignment, secondary structure prediction and our drRecJ structures (*Figure 1—figure supplement 1*), we provide a possible mechanism of CDC45 binding ssDNA. CDC45 exhibits sequence homology to the drRecJ DHH domain, which possesses five conserved motifs. However, three key residues (Asp79, Asp135 and His159 in drRecJ) that coordinate catalytic metal ions are not conserved in CDC45, which is consistent with the inactivation and the absence of metal ion in purified CDC45 proteins (*Krastanova et al., 2012*). Second, residues that compose the terminal 5′-phosphate-binding pocket (Arg109, Ser371 and Arg373 in drRecJ) are missing in CDC45, which may explain the opposite 3′-5′ DNA binding polarity of CDC45 compared with RecJ (*Szambowska et al., 2014*). Moreover, DNA binding for RecJ is metal ion independent (*Figure 3—figure supplement 2*) (*Han et al., 2006*). The residues involved in protein-DNA stacking interactions (Tyr114, Val224 and Phe269 in drRecJ) and the helical gateway (Arg280, R313, R314 and Lys353 in drRecJ) are conserved in CDC45, which suggests that the DNA binding of CDC45 is possibly similar to that of RecJ.

## RecJ substrate recognition, processivity and catalysis

The structures presented in this study reveal an elegant mechanism for how RecJ binding to DNA initiates DSB end resection in a 5′-3′ direction. First, the terminal of ssDNA is anchored to the 5′-phosphate binding pocket above the active site, which explains the RecJ hydrolyzing DNA with 5′-3′ polarity. This terminal 5′-phosphate-binding pocket appears to be shared by other 5′-3′ exonucleases such as the λ exonuclease (*Zhang et al., 2011*), XrnI (*Jinek et al., 2011*) and RNaseJ (*Zhao et al., 2015*). Second, the active site encloses a single deoxyribonucleioside, which renders RecJ an exonuclease discriminating DNA over RNA. Third, the entrance to the RecJ active site is guarded by the helical gateway, which prevents dsDNA from entering the active site. Fourth, a number of residues form stacking interactions with every two DNA bases. Biochemically, the end resection by the RecJ nuclease is processive. Alanine substitution of these residues resulted in multiple intermediate digestion bands, which indicates that the protein-DNA staking interactions are required for the mechanism of processivity of the RecJ nuclease. Fifth, the OB fold domain is located on the side of the DNA entrance, which positions the downstream DNA towards the active site.

It has recently been shown that RecJ alone is able to process DNA with a 5′-ssDNA overhang (*Morimatsu and Kowalczykowski, 2014*). Mechanistically, RecJ is required to locally melt dsDNA and bind to the consequential 5′-ssDNA for the resection. Our analyses suggest that both the nuclease core and the OB fold domain contribute to the RecJ resection of DNA that bears a 5′-ssDNA overhang. In addition to the reduced nuclease activity and processivity, the mutation of the key residues involved in nuclease core-DNA interactions showed strong reaction-stops at the ss-dsDNA

junction (*Figure 3—figure supplement 1C*), which suggests that the complete processivity is essential for the RecJ resection of DNA with a 5′-ssDNA overhang. On the other hand, the OB fold domain is located at the ss-dsDNA junction. Ala substitution of Tyr496 impaired the nuclease activity against DNA with 5′-ssDNA overhang and ssDNA, but to a lesser extent, which indicates that the OB fold domain is also critical for this type of end resection. Interestingly, stacking interaction between Tyr496 and a purine base appears to be preferred, which may also be important for the dsDNA melting.

Our drRecJ:DNA and drRecJ:dTMP structures suggest that RecJ nucleases employ the two-metal-ion catalysis observed for many nucleic acid processing enzymes (*Nakamura et al., 2012*; *Yang, 2011*; *Zhao et al., 2013*; *2015*). In summary, after deprotonation, a water molecule serves as the nucleophile poised on the 5′-side to attack the scissile phosphate. After bond breakage, His397, which is located at the opposite side of the nucleophilic attack, stabilizes the 3′-O leaving group. The resultant mononucleotide and one or both metal ions can be directly exported through the exit tunnel (Arg393). The newly exposed 5′-phosphate, which is generated in each cycle of the exonuclease reaction, further translocates into the terminal 5′-phosphate-binding pocket, making the next nucleotide ready for the cleavage. RecJ is able to digest DNA that contains abasic sites (*Dianov et al., 1994*); this binding pocket is located right above the active site, which indicates that the attraction of 5′-phosphate to this binding pocket helps to accurately position the scissile phosphodiester bond of the DNA substrate in the active site.

## Implication for DSB repair in *D. radiodurans*

*D. radiodurans* is capable of surviving high doses of ionizing radiation, which shatters its genome into several hundred fragments. The resulting numerous DSBs can be rapidly repaired by its super-efficient RecA-mediated DNA repair system (*Bentchikou et al., 2010*; *Slade et al., 2009*). The functional RecF pathway is required for the RecA-mediated DSB repair in vivo, as *D. radiodurans* naturally lacks RecB and RecC enzymes. *E. coli* has a minimum of three 5′-3′ exonucleases (RecJ, RecBCD and ExoVII), whereas RecJ is the only 5′-3′ exonuclease in *D. radiodurans*, which plays an essential role in DSB end resection (*Lovett, 2011*). In contrast to the use of $Mg^{2+}$ for DNA digestion by ecRecJ, $Mn^{2+}$ appears to be preferred for drRecJ catalysis. This metal-ion preference may be attributed to the following two properties: First, the ionic radius of $Mn^{2+}$ is similar to that of $Mg^{2+}$ but with less-rigid coordination requirements (*Harding, 2000*), making it suitable for His159 coordination. In fact, only one $Mg^{2+}$ was observed in crystal form that contains $Mg^{2+}$ instead of $Mn^{2+}$ (unpublished data), which is consistent with the extremely low drRecJ nuclease activity in the presence of $Mg^{2+}$ (*Figure 1—figure supplement 2*). Second, Daly and co-workers noted that *D. radiodurans* contained about 300 times more intracellular $Mn^{2+}$ than that in *E. coli*, which plays a critical role in reactive oxygen species detoxification and protein protection (*Daly et al., 2004*; *Daly, 2009*). Thus, the $Mn^{2+}$ preference of drRecJ may also be an example of the evolution adaptation. Indeed, in *D. radiodurans*, many enzymes involved in DNA repair (e.g., PolI, PolX) prefer to use $Mn^{2+}$ for catalysis (*Cheng et al., 2015a*; *Heinz and Marx, 2007*; *Lecointe et al., 2004*; *Zhang et al., 2014*; *Zhao et al., 2015*), which indicates that the accumulation of $Mn^{2+}$ may also contribute to the robust DSB repair.

Importantly for the efficient HR, the DSB with different end structures must be promptly processed. Our structures and biochemical data are consistent with the helicase-nuclease coordination end resection mechanism (*Figure 7C*) and suggest that the extended C-terminal domain of drRecJ, which is absent in *E. coli*, is critical for its efficient end resection in *D. radiodurans*. The C-terminal domain can enhance the nuclease activity of drRecJ in vitro (*Figure 2C*), which suggests that drRecJ may exhibit a more effective nuclease activity in *D. radiodurans*. In addition to ssDNA, drRecJ alone is able to initiate DNA end resection with 5′-ssDNA overhangs processively. Interestingly, drSSB stimulates drRecJ resection of DNA that bears a 3′-ssDNA overhang to a certain extent in vitro, possibly via the direct interaction between the drRecJ C-terminal domain and SSB-Ct, which suggests that drRecJ together with drSSB might be capable of processing this type of DSB end structure in vivo. On the other hand, SSB in *E. coli* recruits RecO protein to ssDNA through SSB-Ct. drRecO does not interact with SSB-Ct (*Ryzhikov et al., 2011*), which suggests the existence of alternative pathway of HR initiation in *D. radiodurans*. drRecQ further stimulates the reaction, which indicates that the 3′-5′ helicase activity is also required for the efficient end resection. However, we found no evidence for the direct interaction between drRecJ and drRecQ, which suggests that other

redundant helicases (e.g., UvrD) may also assume the unwinding responsibilities in vivo (*Bentchikou et al., 2010*; *Ithurbide et al., 2015*).

Collectively, we report the crystal structure of drRecJ in complex with SSB-Ct peptide and a DNA substrate representing a DSB end substrate, with 5′-ssDNA overhang placed within the active site, where DSB end resection occurs. These observations reveal the shared and distinctive features of DHH family proteins and enable us to propose a unified mechanism for substrate recognition and the exonucleolytic cleavage activity for RecJ family nucleases. The novel C-terminal domain involved in protein-protein interaction suggests a more effective DSB end resection in *D. radiodurans*.

## Materials and methods

### Cloning and strain constructions

Full-length (residues 1-705aa), nuclease-core (drRecJ$_{core}$, residues 48-431aa), C-terminal domain truncation (drRecJ$_{\Delta C}$, residues 1-531aa), and C-terminal α helix truncation (drRecJ$_{\Delta\alpha-CT}$, residues 1-690aa) of drRecJ were amplified by PCR and cloned to the modified pET28a expression vector, pET28-HMT, which contains a fused N-terminal 6×His-tag, a MBP-tag and a TEV protease recognition sequence (His-MBP-TEV) as described (*Austin et al., 2009*). Full-length drSSB (residues 1-301aa), C-terminal truncated drSSB (drSSB$_{\Delta C}$, residues 1-293aa), and full-length drRecQ (residues 1-824aa) were also cloned to pET28-HMT expression vector. All the constructed expression vectors were transformed into *Escherichia coli* Rossetta (DE3) strain. For phenotypic assays, *drrecJΔCα* was also cloned to the shuttle vector pRADK, named as pk-*recJΔCα*. *drrecJ* complemented strain (*ΔrecJ/pk-recJΔCα*) was constructed by transforming pk-*recJΔCα* into *drrecJ* knockout strain (*ΔrecJ*) as described previously (*Jiao et al., 2012*), followed by sequencing identification. Site directed mutagenesis was performed with a QuikChangeTM Site-Directed Mutagenesis Kit from Stratagene (La Jolla, CA) as described (*Cheng et al., 2015a*). Primers used for cloning and mutageneses are listed in *Supplementary file 1*. All bacterial strains and plasmids used in this study are listed in *Supplementary file 2*.

### Protein preparation

All the drRecJ proteins were expressed and purified in a similar way as reported previously (*Cheng et al., 2015b*; *Jiao et al., 2012*). In brief, transformed *Escherichia coli* Rossetta (DE3) clones were grown at 37°C in LB medium containing 50 µg/ml Kanamycin to an optical density at 600 nm of 0.6–0.8. Protein expression was induced at 30°C for 5 hr by adding isopropyl-β-D-thioga-lactopyranoside (IPTG) with a final concentration of 0.4 mM. After harvesting, cells were resuspended in lysis buffer (20 mM Tris (pH 7.5), 1 M NaCl, 5% (w/v) glycerol, 3 mM β-ME and 10 mM imidazole), lysed by sonication and centrifuged at 15,000 × g for 30 min at 4°C. The supernatant was purified by HisTrap HP column (GE Healthcare), equilibrated with buffer A (20 mM Tris (pH 7.5), 1 M NaCl, 5% (w/v) glycerol and 10 mM imidazole), washed with 30 mM imidazole and finally eluted with 300 mM imidazole. After TEV-tag-removal using TEV protease, the protein was loaded onto the MBPTrap HP column (GE Healthcare, Fairfield, CT) to remove the uncleaved protein. The flow-through fractions were collected and loaded onto a Heparin HP column (GE Healthcare, Fairfield, CT) pre-equilibrated with buffer B (20 mM Tris (pH 7.5), 100 mM NaCl, 1 mM DTT, 5% (w/v) glycerol). Fractions containing drRecJ proteins were eluted with a linear gradient from 100 mM to 500 mM NaCl. The protein was finally purified by Superdex 200 10/300 GL column (GE Healthcare, Fairfield, CT) with buffer C (20 mM Tris (pH 7.5), 100 mM NaCl, 0.1 mM EDTA and 1 mM DTT) and stored at −80°C. drSSB and drSSB$_{\Delta C}$ were expressed similar as drRecJ, and purified as reported previously (*Cheng et al., 2014*). drRecQ was induced at 16°C for 16 hr by adding isopropyl-β-D-thioga-lactopyranoside (IPTG) with a final concentration of 0.4 mM. The purification procedure of drRecQ was similar as drSSB, expect that Superdex 200 column was used instead of Superdex 75 in the final purification step.

### Crystallization and structure determination

Crystallization trials were carried out by the sitting drop vapor diffusion method at 293 K. Fresh purified drRecJ was concentrated to ∼18 mg/ml and centrifuged to remove insoluble fraction before crystallization. After a series of screening tests and optimizations, crystals of RecJ-dTMP (complex I) were obtained in 1.3 M Li$_2$SO$_4$, 100 mM MES (pH 6.5), 2.5 mM MnCl$_2$ and 10 mM dTMP. For RecJ:

DNA complex, protein was mixed with DNA (*Supplementary file 3*) at a 1:1.2 molar ratio and concentrated to ~18 mg/ml. Complex II crystals were grown in 1.5 M $Li_2SO_4$, 100 mM MES (pH 6.5), and 2.5 mM $MnCl_2$. Complex III crystals were grown in similar condition, followed by soaking with 0.2 mg/mlSSB-Ct (EDDLPF) peptide for 24 hr. Cryocooling was achieved by stepwise soaking the crystals in reservoir solution containing 10, 20, and 30% (w/v) glycerol for 3 min and flash freezing in liquid nitrogen. Diffraction intensities were recorded on beamline BL17U at Shanghai Synchrotron Radiation Facility (Shanghai, China) and were integrated and scaled with the XDS suite (*Kabsch, 2010*). The structures were determined by molecular replacement using ttRecJ (2ZXP) as the search model (*McCoy et al., 2007*). Structures were refined using PHENIX (*Adams et al., 2010*) and interspersed with manual model building using COOT (*Emsley et al., 2010*). Later stages of refinement utilized TLS group anisotropic B-factor refinement. The refined model contained one drRecJ molecule in the asymmetric unit. Two catalytic $Mn^{2+}$ ions and dTMP (complex I), DNA (complex II) or DNA/SSBct (complex III) were observed. The statistics for data collection and refinement are listed in *Table 1*. All the residues are in the most favorable and allowed regions of the Ramachandran plot. All structural figures were rendered in PyMOL (www.pymol.org).

## Nuclease activity

All the oligo DNA and RNA were purchased from Sangon (Shanghai, China) and Takara (Dalian, China) with 3′-end labeled by 6-carboxyfluorescein (6-FAM). The sequences of oligos were listed in the *Supplementary file 3*. DNA with 3′- (KY06, 6 nt overhang) or 5′-overhang (KY04, 14 nt overhang) ssDNA was obtained by annealing in annealing buffer (10 mM HEPES (pH 8.0), 50 mM NaCl, 0.1 mM EDTA) by heating (95°C) for 5 min and slow cooling to 4°C. KY03-KY07 were used to determine the drRecJ substrate preference (*Figure 1B*). KY08 was used to test the nuclease activities of different truncations of drRecJ (*Figure 2C*). KY03 was used to test the nuclease activities of mutant drRecJ proteins (*Figure 3E* and *5C*). KY03 and KY04 were used to test the nuclease activity of drRecJ on ssDNA and DNA with 5′-ssDNA overhangs (*Figure 4C* and *Figure 3—figure supplement 1C*). KY08 and KY09 were used to test the different digestion efficiency of drRecJ on poly (dA) and poly (dT). KY06 was used to test the stimulation of drRecJ nuclease activities by drSSB and drRecQ (*Figure 6C* and *7B*). For a typical nuclease assay, 100 nM DNA was incubated with various concentrations (2.5–500 nM) of freshly prepared full-length, truncated or mutant drRecJ proteins in a 10 μl reaction volume containing 50 mM Tris (pH 7.5), 60 mM KCl, 0.1 mg/ml BSA, 1 mM DTT, 5% (v/v) glycerol and 0.1 mM $MnCl_2$ at 30°C for 20–30 min, in the presence or absence of 100–200 nM drSSB or drRecQ proteins. The reactions were stopped with 2×stop buffer (10 mM EDTA, 98% formamide), incubating at 95°C for 10 min and flash cooled on ice. Reaction products were resolved on 12–20% polyacrylamide gels containing 7 M urea. The gels were imaged at fluorescence mode (FAM) on Typhoon FLA 9500 (GE), and bands were analyzed using Image J Software (National Institutes of Health, USA). To determine the metal preference, KY08 was used and $MnCl_2$ (0.01–10 mM), $MgCl_2$ (0.01–10 mM) or EDTA (10 mM) was added in the reaction buffer. Additional $Mg^{2+}$ (2 mM) and ATP (2 mM) were present in the reaction buffer when drRecQ was needed. For steady state measurements, typically 5–10 nM drRecJ were incubated with saturated substrate (KY03, 0–2 μM) at 30°C for 20 min. All reactions were independently repeated at least three times. The $k_{cat}$ and $K_M$ were derived from generalized nonlinear least-squares using Michaelis-Menten equation, from which the apparent second order rate constant ($k_{cat}/K_M$) was determined from a plot of normalized initial rate ($v$/[E]) versus substrate concentration ([S]). Kinetic parameters of wild-type and mutant drRecJ proteins can be found in *Table 2*.

## Electrophoretic mobility shift assays

100 nM 3′-labeled ssDNA (KY08) was incubated with different concentrations of RecJ (31.5, 62.5, 125, 250, 500, 1000 and 2000 nM) in a 20 μl reaction volume containing 50 mM Tris (pH 7.5), 60 mM KCl, 0.1 mg/ml BSA, 1 mM DTT and 5% (v/v) glycerol at 30°C for 20 min. Samples were separated on 8% native polyacrylamide gels in 1 × TBE buffer. The gels were imaged at fluorescence mode (FAM) on Typhoon FLA 9500 (GE). The bands were analyzed using Image J Software (National Institutes of Health, USA) and the graph was created by Graphpad Prism 6 Software.

## Phenotypic assay

*D. radiodurans* wild type strain R1 and its derivatives were grown at 30°C either in TGY broth (0.5% tryptone, 0.1% glucose, 0.3% yeast extract) or on TGY agar plate (with 1.25% agar). Phenotypic assays were performed as described previously (*Cheng et al., 2015b*). Cells were grown to early exponential phase (OD600 = 0.6-0.8) and incubated with 20 mg/ml of MMC at 30°C for 20 min, diluted to various concentrations and dotted onto TGY plates. Plates were cultured at 30°C for 2–3 days. The cells without MMC treatment were set as control. For temperature-dependent assay, a plate was cultured at 37°C for 2–3 days.

## Acknowledgement

We would like to thank the staff at the Shanghai Synchrotron Radiation Facility (SSRF in China) for assistance in data collection. We thank Dr. David Waugh (National Cancer Institute) for the generous gift of the protein expression vector pET28a-MBP-TEV. We also thank Dr. Wei Yang for the critical reading of the manuscript. This work was supported by National Basic Research Program of China (2015CB910600), the Zhejiang Provincial Natural Science Foundation for Outstanding Young Scientists (LR16C050002), grants from the National Natural Science Foundation of China (31500656, 31570058, 31370102, 31210103904), and a special Fund for Agroscientific Research in the Public Interest (201103007).

## Additional information

### Funding

| Funder | Grant reference number | Author |
|---|---|---|
| Ministry of Science and Technology of the People's Republic of China | 2015CB910600 | Hong Xu<br>Ye Zhao<br>Yuejin Hua |
| National Natural Science Foundation of China | 31570058 | Liangyan Wang |
| National Natural Science Foundation of China | 31500656 | Ye Zhao |
| Natural Science Foundation of Zhejiang Province | LR16C050002 | Ye Zhao |
| National Natural Science Foundation of China | 31210103904 | Yuejin Hua |
| National Natural Science Foundation of China | 31370102 | Yuejin Hua |

The funders had no role in study design, data collection and interpretation, or the decision to submit the work for publication.

### Author contributions

KC, Designed the experiments, Wrote the paper, Carried out protein purification, crystallization and nuclease assays, Acquisition of data, Analysis and interpretation of data; HX, XC, Did point mutations and expressed the proteins, Acquisition of data, Drafting or revising the article; LW, BT, Performed the phenotypic assays, Acquisition of data, Drafting or revising the article; YZ, Designed the experiments, Wrote the paper, Determined and analyzed the structures, Acquisition of data; YH, Designed the experiments, Wrote the paper

### Author ORCIDs

Ye Zhao, http://orcid.org/0000-0002-5455-2586

## Additional files

**Supplementary files**
• Supplementary file 1. Primers used for cloning and mutagenesis.

• Supplementary file 2. Strains and plasmids used in this study.

• Supplementary file 3. DNA substrates used in this study.

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
