## [Decision Letter]

Thank you for submitting your work entitled "Structural basis for DNA 5´-end resection by RecJ" for consideration by *eLife*. Your article has been reviewed by two peer reviewers, and the evaluation has been overseen by Stephen Kowalczykowski as the Reviewing Editor and John Kuriyan as the Senior Editor.

The reviewers have discussed the reviews with one another and the Reviewing Editor has drafted this decision to help you prepare a revised submission.

The following individuals involved in review of your submission have agreed to reveal their identity: Susan Lovett (peer reviewer); and Sergey Korolev (peer reviewer).

Summary:

This manuscript presents a new structural study of the RecJ exonuclease of *Deinococcus radiodurans*. This enzyme plays a critical role in the RecFOR pathway of recombination and is present in almost all eubacteria. It is a member of a larger group that includes RNases, phosphatases and the Cdc45 replication protein of eukaryotes. Although a structure of a partial RecJ was presented earlier for *Thermus thermophilus* (Yamagata et al. 2002), this new structure encompasses the entire protein, with co-crystals with dTMP and ssDNA provides a rationale for several of its properties including 5' to 3' polarity, DNA binding, ssDNA specificity, processivity and interactions with SSB. The data are well presented, the figures are clear and informative. A number of mutations in key residues support the conclusion of the paper.

Essential revisions:

The authors need to explicitly describe tails length in main text.

An experiment that is not reported, regarding the SSB-RecJ interaction, is to simply use linear ssDNA in the presence of WT or Y575A RecJ and WT or δ-C SSB. This experiment would determine whether the SSB-RecJ interaction is required for RecJ to resect SSB-covered ssDNA.

Figure 7: This experiment is not very good. Although the authors showed that both SSB and RecQ are required for RecJ to resect 3'-overhang DNA in Figure 7, this result is not consistent with Figure 6, where SSB alone stimulated RecJ-mediated resection of 3'-overhang DNA (lanes 4 and 5). The authors should clarify this apparent discrepancy and/or repeat these experiments.

Reviewer #1:

I have only a few comments to improve the paper.

Introduction: Although the RecFOR pathway can mediate recombination at double-strand breaks (DSBs), in *E. coli* at least it mediates recombination at ssDNA gaps in DNA. The interest is on DSB repair in *Deinococcus*, because of its radioresistance; however gap recombination should be mentioned briefly.

Introduction, first paragraph: the Han et al. 2006 paper was the first to show SSB stimulation of DNA binding and hydrolysis for the *E. coli* protein.

Introduction, fourth paragraph: The Yamagata 2002 structure only showed one metal ion, Mn^2+^ (I re-checked this to be sure.) This should be corrected. Sutera et al. 1999 (J. Bacteriol 181: 6098) mutated the equivalents of D79, D81, D83, D158, H159, H160, D233, R373, and H397 in *E. coli* RecJ and showed that these are defective in vivo and in vitro, which should be cited. Interestingly, some of these are genetically dominant and the explanation was that they bind, but do not degrade DNA. I was hoping that these dominant residues would define one of the two metal binding sites but this does not appear to be the case.

Subsection “Nucleotide binding site and the catalytic mechanism”: Try80, Try114 should be Tyr80, Try114. These residues are not conserved: Tyr80 is Phe in *E. coli* and other proteobacteria and the 114 equivalent can be His in some other bacteria. How would this affect the nucleotide binding site?

Discussion, third paragraph: DNA binding for *E. coli* RecJ is Mg^2+^ independent (again, Han et al. 2006), so this helps explain how Cdc45 can employ similar DNA binding without the catalytic site.

Reviewer #2:

The manuscript "Structural basis for DNA 5´-end resection by RecJ" describes structural and biochemical studies of *D. radiodurans* RecJ exonuclease.

Experiments are of exceptional quality and are well presented. The results are quite interesting and have a significant impact in the field of DNA replication and repair and are worth publishing.

In addition to excellent structural studies of the RecJ cleavage mechanism, the authors revealed several interesting biochemical properties, e.g. stimulation of 3' overhang substrate cleavage by SSB without helicase involvement.

---

## [Author Response]

*Summary: This manuscript presents a new structural study of the RecJ exonuclease of Deinococcus radiodurans. This enzyme plays a critical role in the RecFOR pathway of recombination and is present in almost all eubacteria. It is a member of a larger group that includes RNases, phosphatases and the Cdc45 replication protein of eukaryotes. Although a structure of a partial RecJ was presented earlier for Thermus thermophilus (Yamagata* et al.

*2002), this new structure encompasses the entire protein, with co-crystals with dTMP and ssDNA provides a rationale for several of its properties including 5' to 3' polarity, DNA binding, ssDNA specificity, processivity and interactions with SSB. The data are well presented, the figures are clear and informative. A number of mutations in key residues support the conclusion of the paper. Essential revisions: The authors need to explicitly describe tails length in main text.*

Yes, a mention to the tails length of 5′ overhang DNA (14 nt) and 3′ overhang DNA (6 nt) was made in the revised version text (main text and Methods section).

An experiment that is not reported, regarding the SSB-RecJ interaction, is to simply use linear ssDNA in the presence of WT or Y575A RecJ and WT or δ-C SSB. This experiment would determine whether the SSB-RecJ interaction is required for RecJ to resect SSB-covered ssDNA.

Thank you for the suggestion. We have tested the resection of SSB-coated ssDNA by WT/Y575A RecJ and WT/Δ-C SSB (please see gel below, added as Figure 6—figure supplement 1 in revised version). For wild-type drRecJ, this nuclease activity is enhanced by SSB_WT_ (lanes 4-5) but not by SSB_ΔC_ (lanes 6-7). In contrast, neither SSB_WT_ (lanes 10-11) nor SSB_ΔC_ (lanes 12-13) substantially stimulate the RecJ_575_ resection of ssDNA to SSB_WT_ level. These results are consistent with the drRecJ resection of DNA with a 3′-ssDNA overhang, which indicates that the SSB-RecJ interaction is also required for RecJ to digest SSB-covered ssDNA. A mention to this was made in the revised version text. “For ssDNA, the stimulation is similar to that of the resection of the DNA with a 3´-ssDNA overhang, suggesting that the SSB-RecJ interaction is required for RecJ to resect SSB-covered ssDNA (Figure 6—figure supplement 1).” The Methods section has been updated as well.

Figure 7: This experiment is not very good. Although the authors showed that both SSB and RecQ are required for RecJ to resect 3'-overhang DNA in Figure 7, this result is not consistent with Figure 6, where SSB alone stimulated RecJ-mediated resection of 3'-overhang DNA (lanes 4 and 5). The authors should clarify this apparent discrepancy and/or repeat these experiments.

The “discrepancy” is caused by different concentrations of SSB protein used in the assays (200 nM for Figure 6 and 100 nM for Figure 7), which have been mentioned in the figure legends. Actually, the stimulation of drRecJ-mediated resection of 3′-overhang DNA depends on the SSB concentrations (please see gel below) that high concentrations of SSB protein could drastically stimulate this type of the resection even in the absence of RecQ protein. We think the possible explanation is that high concentrations of drSSB could remove DNA secondary structures (dsDNA) and facilitate drRecJ resection. So, we decided to use relatively low concentration of SSB to conduct the nuclease assays of Figure 7 to check the RecQ-effect on the resection. So, in the revised Figure 6 and Figure 7, the concentrations of SSB were labeled to clarify the different amount of SSB used in the assays. Thank you for this good comment.

*Reviewer #1: I have only a few comments to improve the paper. Introduction: Although the RecFOR pathway can mediate recombination at double-strand breaks (DSBs), in E. coli at least it mediates recombination at ssDNA gaps in DNA. The interest is on DSB repair in Deinococcus, because of its radioresistance; however gap recombination should be mentioned briefly.*

Thank you for the suggestion. A mention to this was made in the revised version text and one more reference (Lovett et,al., 1989, PNAS) was cited. “In addition to HR, RecJ is involved in ssDNA gap repair, base excision repair and methyl-directed mismatch repair (Burdett et al., 2001; Dianov et al., 1994; Lovett and Kolodner, 1989).”

*Introduction, first paragraph: the Han* et al. *2006 paper was the first to show SSB stimulation of DNA binding and hydrolysis for the E. coli protein.*

Sorry for missing such an important reference and it has been cited in the revised manuscript

*Introduction, fourth paragraph: The Yamagata 2002 structure only showed one metal ion, Mn^2+^ (I re-checked this to be sure.) This should be corrected. Sutera* et al. *1999 (J. Bacteriol 181: 6098) mutated the equivalents of D79, D81, D83, D158, H159, H160, D233, R373, and H397 in E. coli RecJ and showed that these are defective* in vivo *and* in vitro*, which should be cited. Interestingly, some of these are genetically dominant and the explanation was that they bind, but do not degrade DNA. I was hoping that these dominant residues would define one of the two metal binding sites but this does not appear to be the case.*

Yes, the 2002 PNAS paper only showed one metal ion. However, the second paper published in JBC (Wakamatsu, 2010) observed both two metal ions in the structure.

The reference about the mutagenesis of conserved motifs by Sutera (J. Bacteriol 181: 6098) was cited in the revised manuscript (subsection “Nucleotide binding site and the catalytic mechanism”). Indeed, these residues are involved in metal ion binding (D79, D81, D83, H159, D223), catalysis (D158, H160, H397) and 5′-phosphate binding (R373). We agree with your point that these mutations mainly impair the RecJ cleavage and the protein can still bind DNA. A mention to this was made in the revised version text (same section). “[…] which is consistent with the results of equivalent mutations in ecRecJ (Sutera et al., 1999).”

*Subsection “Nucleotide binding site and the catalytic mechanism”: Try80, Try114 should be Tyr80, Try114. These residues are not conserved: Tyr80 is Phe in E. coli and other proteobacteria and the 114 equivalent can be His in some other bacteria. How would this affect the nucleotide binding site?*

We are sorry for the typo and have corrected all the errors in the revised manuscript. For Tyr80 and Tyr114, it is true that these two residues, as you mentioned, are not strictly conserved in all the bacteria. However, the side chains of these residues (Tyr, Phe, His) all contain aromatic rings, which form π stacking interactions (base stacking interactions) with substrate DNA. Thus, these residues have similar side-chain structure and conserved functions, which are critical for the nucleotide binding. In the revised main text, we have deleted the word “conserved”

*Discussion, third paragraph: DNA binding for E. coli RecJ is Mg^2+^ independent (again, Han* et al. *2006), so this helps explain how Cdc45 can employ similar DNA binding without the catalytic site.*

Yes, A mention to this was made in the revised version text with this reference cited. “Moreover, DNA binding for RecJ is metal ion independent (Figure 3—figure supplement 2) (Han et al., 2006).”